# Inter-valley coherent order and isospin fluctuation mediated superconductivity in rhombohedral trilayer graphene

Shubhayu Chatterjee [1] ✉, Taige Wang[1,2], Erez Berg [3] & Michael P. Zaletel[1,2]

Superconductivity was recently discovered in rhombohedral trilayer graphene (RTG) in the absence of a moiré potential. Superconductivity is observed proximate to a metallic state with reduced isospin symmetry, but it remains unknown whether this is a coincidence or a key ingredient for super-conductivity. Using a Hartree-Fock analysis and constraints from experiments, we argue that the symmetry breaking is inter-valley coherent (IVC) in nature. We evaluate IVC fluctuations as a possible pairing glue, and find that they lead to chiral unconventional superconductivity when the fluctuations are strong. We further elucidate how the inter-valley Hund's coupling determines the spin-structure of the IVC ground state and breaks the degeneracy between spin-singlet and triplet superconductivity. Remarkably, if the normal state is spin-unpolarized, we find that a ferromagnetic Hund's coupling favors spin-singlet superconductivity, in agreement with experiments. Instead, if the normal state is spin-polarized, then IVC fluctuations lead to spin-triplet pairing.

The experimental discovery of robust superconductivity in graphene-based moiré heterostructures has placed graphene in the spotlight for studying the physics of strong electronic correlations[1–8]. Very recently, superconductivity was observed in an even simpler system − ABC-stacked rhombohedral trilayer graphene (RTG) *without any moiré pattern*[9]. Near charge neutrality, just like monolayer graphene, the low-energy electrons of RTG are characterized by an isospin index that includes valley and spin[10,11]. Superconductivity emerges on the cusp of isospin symmetry breaking transitions in hole-doped RTG in the pre-sence of a perpendicular displacement field. In particular, there are two superconducting phases (referred to as SC1 and SC2 in Ref. [9]) that flank two distinct isospin symmetry-broken phases [called a 'partially isospin polarized' (PIP) phase in Ref. [9]]. While SC1 is suppressed by in-plane Zeeman fields and respects the Pauli paramagnetic limit[12,13], SC2 appears to strongly violate this limit. Further, the low level of disorder in the sample, as evidenced by $\mu$m-scale mean-free path of electrons, leaves open the possibility for unconventional superconductors.

These remarkable observations naturally lead to important questions. What is the nature of isospin symmetry-breaking in the metallic phases of RTG? What are the pairing symmetries of SC1 and

SC2 that emerge on the verge of isospin symmetry-breaking? What role, if any, do electronic correlations play in aiding or suppressing superconductivity?

In this paper, we propose isospin fluctuations as an all-electronic mechanism of superconductivity in RTG. We first argue that the experimental data strongly constrains the nature of spontaneous symmetry-breaking in the correlated metallic states. In particular, we demonstrate using self-consistent Hartree-Fock calculations that a promising candidate state near SC1 is an inter-valley coherent (IVC) metal that spontaneously breaks the $U(1)_v$ valley conservation sym-metry, but lacks net spin or valley-polarization. Depending on the sign of the inter-valley Hund's coupling, such an IVC metal is either a time-reversal symmetric spin-singlet charge-density wave (CDW), or a col-linear spin-density wave (SDW) that breaks time-reversal and global spin-rotation symmetry: both triple the unit cell[14,15]. Near SC2, we propose that a spin-polarized IVC state, which microscopically corre-sponds to a ferromagnetic CDW, may be realized.

Next, we investigate superconducting instabilities that arise from fluctuations of the IVC order parameter. Interestingly, we find that the leading superconducting instability, as determined by solving a mean-

[1]Department of Physics, University of California, Berkeley, CA 94720, USA. [2]Materials Sciences Division, Lawrence Berkeley National Laboratory, Berkeley, CA 94720, USA. [3]Department of Condensed Matter Physics, Weizmann Institute of Science, Rehovot 76100, Israel. ✉e-mail: shubhayuchatterjee@berkeley.edu

field Bardeen-Cooper-Schrieffer (BCS) gap equation, shows a transition as a function of the IVC correlation length $\xi_{IVC}$. At large $\xi_{IVC}$, i.e., closer to criticality, the dominant instability is towards a chiral fully-gapped superconductor, while at smaller $\xi_{IVC}$ the dominant instability is towards a non-chiral nodal superconductor. Because of the presence of an additional valley degree of freedom, both these states could either be spin-singlet or triplet. Within a model accounting only for intra-valley Coulomb scattering, spin-singlet and triplet superconductors are degenerate due to an enhanced $SU(2)_+ \times SU(2)_-$ spin-rotation symmetry (valleys labeled by ±). However, we argue that the inter-valley Hund's coupling arising from lattice-scale effects determines the spin-structure. The existence of valley-unpolarized, spin-polarized phases in RTG implies that the Hund's coupling is ferromagnetic. Remarkably, we find that such a Hund's coupling prefers a *spin-singlet* superconductor, consistent with SC1. In contrast, SC2 is likely a non-unitary spin-triplet which inherits the spin-polarization of the ferromagnetic normal state.

The rest of this paper is organized as follows. In Section "Hamiltonian and symmetries", we introduce the interacting Hamiltonian for RTG and its symmetries. In Section "Inter-valley coherent order", we argue in favor of an IVC phase near SC1 using both Hartree-Fock and analytical calculations, and discuss its real-space and momentum space structures. In Section "Hund's coupling", we discuss how the inter-valley Hund's coupling has an unusual form which favors spin-triplet IVC over spin-polarization when ferromagnetic. In Section "IVC fluctuation mediated superconductivity", we analyze superconducting instabilities arising from IVC fluctuations, and study the role of the Hund's term in splitting the degeneracy between spin-

singlet and triplet superconductors. We conclude in Section "Discussion" with a summary of our main results, comparison to experimental data and recent theoretical work, and an outlook.

## Results

### Hamiltonian and symmetries

ABC-stacked RTG is most accurately described using a six-band model per valley ($K/K'$) and spin[10,11]. All numerical calculations presented in this work use the six-band model with tight-binding parameters taken from Ref. [16] (See Supplementary Material for further details). However, it is useful to develop some intuition for the band structure within an approximate 2-band model which describes the low-energy physics in each valley. The wave-functions of the two bands closest to the Fermi level reside mostly on the non-dimerized sites on the top/bottom layer (denoted by $\sigma = A_1/B_3$ respectively, see Fig. 1a). In this pseudospin basis, the effective Hamiltonian can be written as:

$$H = \sum_{\tau,s,\mathbf{k}} c^\dagger_{\tau,s,\mathbf{k},\sigma} \left( [h_\tau(\mathbf{k})]_{\sigma\sigma'} - \mu\,\delta_{\sigma\sigma'} \right) c_{\tau,s,\mathbf{k},\sigma'} + H_C,$$

$$[h_\tau(\mathbf{k})]_{\sigma\sigma'} = \begin{pmatrix} -u & \frac{v_0^3}{\gamma_1^2}\Pi^3 + \frac{\gamma_2}{2} \\ \frac{v_0^3}{\gamma_1^2}(\Pi^*)^3 + \frac{\gamma_2}{2} & u \end{pmatrix}_{\sigma\sigma'} \quad (1)$$

where $\Pi = \tau k_x + i k_y$, $\tau = \pm$ denotes valley, $s = \uparrow/\downarrow$ labels spin, and $\mu$ is the chemical potential. The band structure parameter $v_0$ is the Dirac velocity of monolayer graphene, $\gamma_1 \sim 300$ meV quantifies the strength of interlayer dimerization, $\gamma_2 \sim -15$ meV is the direct hopping between $A_1/B_3$ that contributes to trigonal warping, and $u \sim 10$s of meV is the

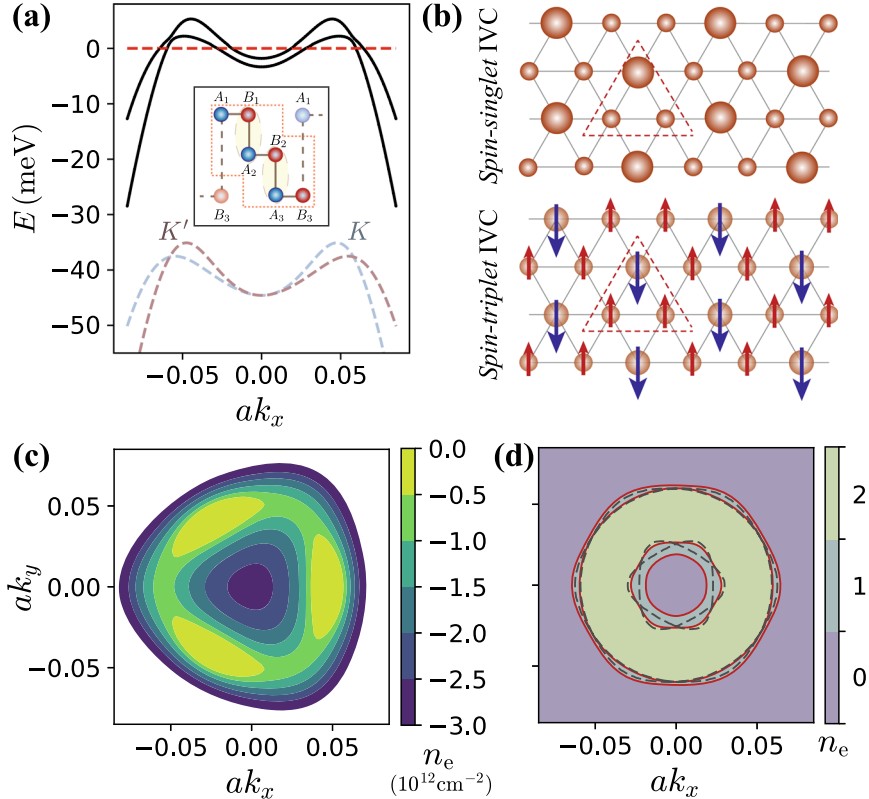

**Fig. 1 | Band structure and real space description of inter-valley coherent (IVC) phases. a** 1D cut of the Hartree-Fock band structure of the for a self-consistent HF IVC state close to the onset of the spin-singlet/triplet IVC phase, with the bare band structure in the two valleys shown by dotted lines below (displaced below for clarity). Inset shows the unit-cell of RTG, with $\{B_i, A_{i+1}\}$ being strongly hybridized ($i=1, 2$) such that the active sublattices $A_1/B_3$ form a triangular lattice. **b** Real space structure of the spin-singlet/CDW IVC (left) and the spin-triplet/SDW IVC (right) on the effective triangular lattice, with dotted lines showing the tripled unit cell in each case. **c** Fermi surface of the single-particle band structure in the $K$ valley at different electron densities $n_e$, assuming fourfold isospin degeneracy. **d** 2D depiction of the reconstructed Fermi surface of the same IVC state as panel **a**, showing two annular pockets. Different colors indicate the number of filled HF bands. The dashed black curves are the Fermi surfaces of HF self-consistent *symmetric* metal at identical filling.

potential difference between the two layers due to the perpendicular electric field. When $\gamma_2 = 0$, the electric field $u$ gaps out the cubic-band touchings, leading to a large density of states (DOS) at the band extrema centered on $K/K'$. Symmetry-breaking is only seen at sufficiently large $u$, presumably because the increased DOS leads to stronger interaction effects[16]. The $\gamma_2$ term then splits the band extrema into three shallow pockets related by $C_3$ rotations about $K/K'$. As shown in Fig. 1c, as the electron density is reduced below neutrality, the topology of the Fermi surface within each valley first transitions from three $C_3$-related pockets to an annulus via a van-Hove singularity, and finally to a distorted disc via a Lifshitz transition. For hole-dopings large enough such that $(v_0 k_F)^3/\gamma_1^2 \gtrsim u, \gamma_2$, the DOS at the Fermi surface is low and interesting interaction effects disappear.

The interacting Hamiltonian $H_C$ is given by:

$$H_C = \frac{1}{2A} \sum_{\mathbf{q}} V_C(\mathbf{q}) : \rho(\mathbf{q})\rho(-\mathbf{q}) : \qquad (2)$$

where $A$ is the sample area, $V_C(\mathbf{q}) = e^2 \tanh(qD)/(2\epsilon q)$ is the repulsive dual gate-screened Coulomb interaction with sample-gate distance $D$, and $\rho(\mathbf{q}) = \sum_{\mathbf{k},\tau,s,\sigma} c^\dagger_{\tau,s,\mathbf{k},\sigma} c_{\tau,s,\mathbf{k}+\mathbf{q},\sigma}$ is the Fourier component of the electron density operator, with $|\mathbf{k}|$ and $|\mathbf{q}|$ being restricted to small values relative to the inverse lattice spacing $a^{-1}$.

The symmetries of $H$ include charge conservation $U(1)_c$, valley-charge conservation $U(1)_v$ generated by $\tau_z$, time-reversal $\mathcal{T}$, translations $T_{1,2}$, mirror reflection $M_x$, and rotation $C_3$. Note there is no inversion symmetry whenever $u \neq 0$, the case of interest, reducing the point group symmetry from $D_{3d}$ to $C_{3v}$[17,18]. The absence of spin-orbit coupling allows us to define a spinless time-reversal $\tilde{\mathcal{T}} = \tau^x K$ which relates dispersions of the $n^{th}$ bands in the two valleys as $\varepsilon_{\tau,n}(\mathbf{k}) = \varepsilon_{-\tau,n}(-\mathbf{k})$. However, trigonal warping splits the valleys locally in momentum space, so $\varepsilon_{\tau,n}(\mathbf{k}) \neq \varepsilon_{-\tau,n}(\mathbf{k})$. Finally, for the interaction defined by $H_C$ there is a separate spin-rotation symmetry in each valley, denoted by $SU(2)_+ \times SU(2)_-$. In reality, this symmetry is broken by lattice-scale effects such as optical phonons and inter-valley Coulomb scattering[19] to a global $SU(2)$ spin rotation; we will return later to the effect of this 'Hund's' coupling $J_H$.

## Inter-valley coherent order

**Isospin symmetry breaking.** We begin by reviewing the experimental constraints on isospin symmetry breaking in the vicinity of SC1[9,16]. Upon approaching charge neutrality from the hole-doped side, a series of phase transitions is observed. The phase transitions are accompanied by Fermi surface reconstruction, visible in quantum oscillations. The first transition is from a fully symmetric phase with fourfold-degenerate annular Fermi surfaces (corresponding to the four isospin degrees of freedom), to a symmetry-broken metallic phase (the PIP phase) with two large and two small Fermi surfaces. The critical density is displacement field (e.g., $u$) dependent, and within our model at $u = 30$ meV, it occurs in the general vicinity of $n_e \sim -1.4 \times 10^{12}$ cm$^{-2}$. The boundary between the two phases is insensitive to an in-plane magnetic field, indicating that the PIP phase is not spin-polarized (this is in contrast to other regions of parameter space, where such dependence is clearly visible). Furthermore, the PIP phase does not have an observable anomalous Hall effect (Andrea F. Young, private communication), which suggests it is time-reversal symmetric. In other regions of the phase diagram, the system is valley-polarized, which produces an experimentally observed anomalous Hall effect due to the valleys' opposing Berry curvature[16], (See Supplementary Material).

The absence of spin and valley polarization suggests that the PIP phase instead has broken $U(1)_v$ symmetry, i.e., it is inter-valley coherent. An alternate possibility would be a spin-valley locked state (SVL) with spins polarized in each valley, but oppositely aligned between the valleys. While such a state is compatible with experiment, we note that it would be disfavored by a ferromagnetic Hund's coupling. As

mentioned above, the presence of nearby spin-polarized, valley-unpolarized phases suggests that the Hund's coupling is ferromagnetic. We shall assume that this is the case, and will not discuss the possibility of a SVL phase further.

In the absence of symmetry breaking, the band dispersion of the two valleys $\varepsilon_\pm(\mathbf{k})$ cross at certain high symmetry points related by $C_3$ and $M_x$. The IVC order hybridizes the valleys, gapping out the band crossings and deforming the $\mathcal{T}$-related annular Fermi surfaces of the two valleys into a small and large annulus, see Fig. 1a, d. We identify this as the "PIP" phase in which quantum oscillations give evidence for a spin-unpolarized state featuring multiple Fermi surfaces with different areas; SC1 lies adjacent to this phase.

To verify that an IVC metal can be energetically favorable, we conduct self-consistent Hartree–Fock (HF) calculations within the six-band model[11]. In these calculations we phenomenologically account for screening from the itinerant electrons by modifying $V_C$ within the Thomas–Fermi approximation, with screening wavevector $q_{TF}$ based on the non-interacting density of states (for details, refer to Supplementary Material). The resulting phase diagram as a function of hole-doping and displacement field is presented in Fig. 2b, and a line cut at a fixed displacement field is shown in Fig. 2a. Over significant regions of hole-doping and displacement fields of 20–40 meV, a spin-unpolarized IVC metal is energetically competitive with the isospin polarized phase (without a Hund's coupling $J_H$, different patterns of isospin polarization, e.g., full spin vs full valley polarization, are degenerate within HF). The precise energetic ordering of the phases depends on details such as $u$ and $q_{TF}$. Nevertheless, we note that the broad features of our phase diagram (Fig. 2b), such as interaction-induced symmetry breaking at large displacement fields, and the phase boundary between the spin-unpolarized IVC metal and the fully symmetric metal, are consistent with experiments.

**Physical description of IVC states.** In the absence of $J_H$ the set of IVC ground states form a degenerate $U(2)$ manifold related by the action of $SU(2)_+ \times SU(2)_-$ spin-rotations[14,15]. Out of this manifold, inter-valley Hund's coupling, as we will elaborate on later, selects either a spin-singlet or triplet IVC. These states have simple real-space structures, as shown in Fig. 1b. The spin-singlet IVC is a $\mathcal{T}$-symmetric CDW at momentum $\mathbf{K} - \mathbf{K}'$, tripling the unit cell. Unlike monolayer and bilayer graphene, where the active sublattices form a honeycomb lattice, in RTG the active sublattices $A_1/B_3$ are stacked vertically, forming a single *triangular* lattice (see Fig. 1a inset). We define the $A_1/B_3$-projected density operator about $\mathbf{K} - \mathbf{K}'$ momentum transfer

$$n_S^{IV}(\mathbf{q}) = \sum_{\mathbf{R}} e^{-i(\mathbf{K}' - \mathbf{K} + \mathbf{q})\cdot\mathbf{R}} \rho(\mathbf{R}), \qquad (3)$$

where $\mathbf{R}$ is the two-dimensional position vector for $A_1/B_3$ sublattices, and $\rho(\mathbf{R}) = \sum_{\sigma = A_1/B_3} \rho_\sigma(\mathbf{R})$ is the total electron density summed over the two sublattices at position $\mathbf{R}$. Thus, we conclude that $n_S^{IV}(\mathbf{q} = 0)$ serves as a complex order parameter for the singlet/CDW IVC. In fact, HF calculations show that the valley off-diagonal part of the self-consistent HF Hamiltonian $H_{HF}$ is very well-approximated by the operator $\Delta_{IVC} n_S^{IV}(0) + h.c.$, where $\Delta_{IVC}$ is the amplitude of the IVC order parameter (see SM (See Supplementary Material), Fig. 3 for a quantitative comparison). Under $C_3$ about an $A_1/B_3$ site, $n_S^{IV}(\mathbf{q}) \rightarrow n_S^{IV}(C_3\mathbf{q})$. Therefore, the IVC order preserves $A_1/B_3$-site centered $C_3$. While a unit-cell tripling would generically be described by a $\mathbb{Z}_3$ order parameter, corresponding to pinning of the $U(1)_v$ phase of the IVC order parameter to one of three distinct values, quartic interactions do not allow for Umklapp terms that break $U(1)_v \rightarrow \mathbb{Z}_3$, such terms appear only at the sextic level[15].

The spin-triplet IVC can be obtained from the singlet one using the $SU(2)_+ \times SU(2)_-$ symmetry, by applying a spin rotation of $\pi$ on one valley relative to the other around an arbitrary axis. The triplet IVC is a

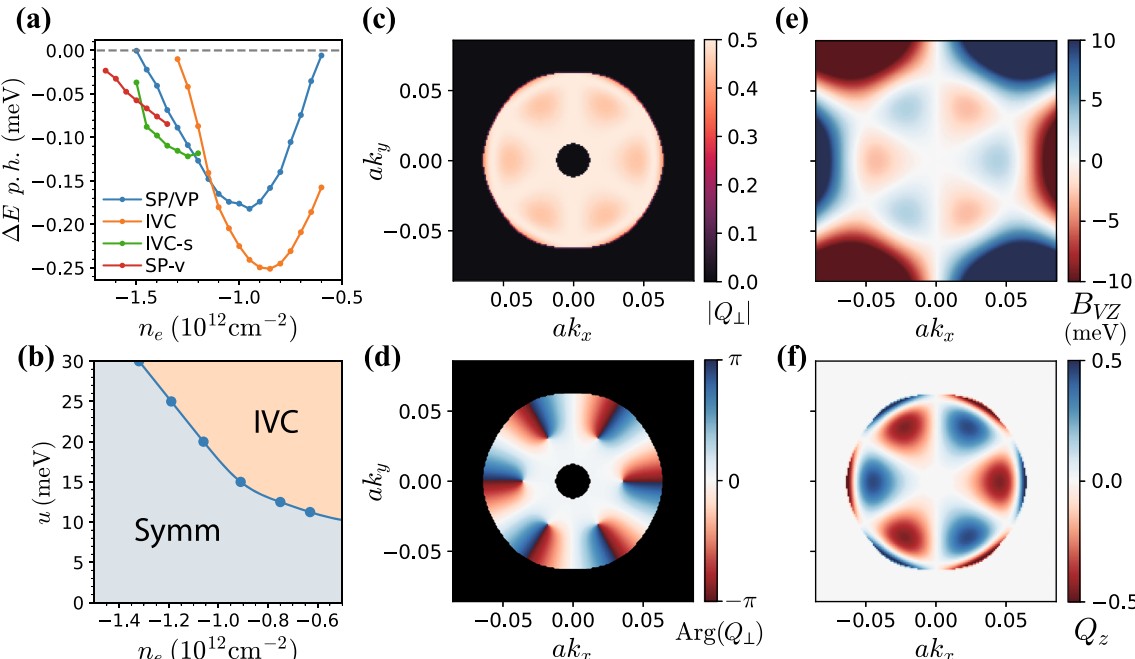

**Fig. 2 | Phase diagram and the IVC order parameter. a** Self-consistent Hartree-Fock energetics of isospin symmetry-broken states for $u = 30$ meV, including (i) spin or valley polarized (SP/VP), (ii) IVC, (iii) partially spin-polarized IVC (IVC-s), and (iv) partially spin and valley-polarized (SP-v) states. (See Supplementary Material for further details of SP-v and IVC-s). All energies are shown in meV per hole, relative to the fully symmetric metal. **b** Hartree-Fock phase diagram as a function of hole-doping $n_e$ and displacement field $u$. Only the fully symmetric metal (Symm) and the spin-unpolarized IVC metal phases have been considered for clarity. **c** Magnitude and **d** phase of the self-consistent HF IVC order $Q_\perp(\mathbf{k}) = Q_x(\mathbf{k}) + iQ_y(\mathbf{k})$ deep in the IVC phase where only the lower IVC band is filled ($n_e = -1.05 \times 10^{12}$ cm$^{-2}$). We have defined $Q_\mu(\mathbf{k}) = \langle\psi^\dagger_{\tau,s,\mathbf{k}}\tau^\mu_{\tau\tau'}\psi_{\tau',s,\mathbf{k}}\rangle$, normalized to unit magnitude. The phase of $Q_\perp(\mathbf{k})$ winds by $12\pi$ around the outer Fermi surface. The region outside the Fermi surface is filled with black for clarity. **f** The valley polarization $Q_z(\mathbf{k})$ in the self-consistent IVC solution follows (**e**) the local valley-Zeeman field $B_{VZ}(\mathbf{k})$.

collinear SDW at momenta $\mathbf{K} - \mathbf{K}'$. In analogy with the singlet IVC, we define the $A_1/B_3$ projected spin-density operator $\mathbf{s}(\mathbf{R}) = \sum_{\sigma = A_1/B_3} \mathbf{s}_\sigma(\mathbf{R})$ about $\mathbf{K} - \mathbf{K}'$ momentum transfer:

$$\mathbf{n}^{\mathrm{IV}}_{\mathrm{T}}(\mathbf{q}) = \sum_{\mathbf{R}} e^{-i(\mathbf{K}'-\mathbf{K}+\mathbf{q})\cdot\mathbf{R}} \mathbf{s}(\mathbf{R}) \tag{4}$$

The spin-triplet IVC parameter is $\mathbf{n}^{\mathrm{IV}}_{\mathrm{T}}(\mathbf{q}=0)$. Thus, the SDW IVC breaks both valley U(1)$_v$ and global SO(3)$_s$ spin-rotation symmetry. Note that a change of the order parameter phase by U(1)$_v$ rotations can be offset by a global spin-rotation about $\hat{\mathbf{n}}$, leading to an order parameter manifold of U(1)$_v \times$ SO(3)/U(1)$_{v+s} \cong$ SO(3)[20-22]. Thus, such a state formally has no long-range or algebraic order at finite temperature[23,24], although it may appear to order at low-enough temperature in finite size systems due to an exponentially diverging correlation length. We also note that within Landau theory, symmetry-allowed couplings between a SDW with momenta $\mathbf{Q}$ and a CDW with momenta $2\mathbf{Q}$ can nucleate such a CDW in presence of long-range SDW order[25]. Thus, the triplet IVC can induce a CDW at $\mathbf{K} - \mathbf{K}'$, which is precisely the singlet IVC order parameter. As such, the strict symmetry distinction between the triplet and singlet IVC is the lack of magnetic order for singlet.

An alternative characterization of the IVC order parameters, useful for studying IVC energetics as well as superconductivity mediated by IVC fluctuations, may be obtained in momentum space. To do so, we use the band-basis, defined via $c^\dagger_{\tau,s,\mathbf{k},\sigma} = \sum_n u^*_{n,\tau,s,\mathbf{k}}(\sigma)\psi^\dagger_{n,\tau,s,\mathbf{k}}$, where $u^*_{n,\tau,s,\mathbf{k}}(\sigma)$ are the Bloch wave-functions and $n$ labels the band index. We define a valence-band projected operator $n^{\mathrm{IV}}_{ss'}(\mathbf{q}) = \sum_{\mathbf{k}} \lambda^{+-}_{\mathbf{q}}(\mathbf{k})\psi^\dagger_{+,s,\mathbf{k}}\psi_{-,s',\mathbf{k}+\mathbf{q}}$, where $\lambda^{+-}_{\mathbf{q}}(\mathbf{k}) = \langle u_{+,s,\mathbf{k}} | u_{-,s,\mathbf{k}+\mathbf{q}}\rangle$ is the inter-valley form factor that captures overlap of wavefunctions from opposite valleys in the valence band, and $U_{ss'}$ is any unitary matrix in spin-space. In this formulation, it is evident that IVC order parameter $n^{\mathrm{IV}}_{ss'}(\mathbf{q}=0)$ lies in the U(2) manifold. This degeneracy is broken by the inter-valley Hund's coupling, which either picks the

spin-singlet CDW with $n^{\mathrm{IV}}_{ss'} \propto \delta_{ss'}$ or the spin-triplet SDW with $n^{\mathrm{IV}}_{ss'} \propto (\hat{\mathbf{n}} \cdot \mathbf{s})_{ss'}$ with an arbitrary unit-vector $\hat{\mathbf{n}}$.

**Energetics of IVC.** We now turn to the energetics of the IVC phase. The IVC order parameter necessarily involves overlap of Bloch states from opposite valleys, and therefore has non-trivial winding originating from opposite chirality of threefold Dirac cones around $K$ and $K'$ points at $u = 0$. The winding of the IVC order parameter in momentum space contributes an additional energy cost relative to an isospin polarized phase (See Supplementary Material). This additional energy cost is responsible for stabilizing an isospin polarized state relative to an IVC state in certain insulators with non-trivial band topology, such as magic angle graphene at certain odd integer filling of flat bands[26-30]. This raises an important question: why, then, is the IVC state energetically favored over a valley-polarized state?

This puzzle can be resolved by noting that an IVC metal can reduce its kinetic energy cost by local valley-polarization[31,32]. To visualize this, it is convenient to think of the IVC order at each $\mathbf{k}$ point as a vector in the x-y plane on the Bloch sphere corresponding to the valley isospin. The trigonal-warping induced kinetic energy mismatch between the valence bands in the $K/K'$ valleys, given by $\varepsilon_+(\mathbf{k}) - \varepsilon_-(\mathbf{k}) = \varepsilon_+(\mathbf{k}) - \varepsilon_+(-\mathbf{k})$, results in a local valley-Zeeman field $B_{VZ}(\mathbf{k})$. The IVC state can thus benefit energetically by canting the valley isospin vector towards $B_{VZ}(\mathbf{k})$ (much like an antiferromagnet gains energy by canting towards an applied magnetic field), without carrying any net valley-polarization as $B_{VZ}(\mathbf{k})$ averages to zero. We explicitly illustrate this energy gain in the supplement (See Supplementary Material), under the approximations of weak IVC order and linearized dispersion close to the Fermi surface. Consistent with this intuition, the self-consistent IVC order parameter obtained from HF also shows local valley-polarization in the vicinity of the Fermi-surface (see Fig. 2c). On the other hand, a valley-polarized phase (corresponding to a vector polarized along $\hat{z}$ on the Bloch sphere) cannot benefit from this local

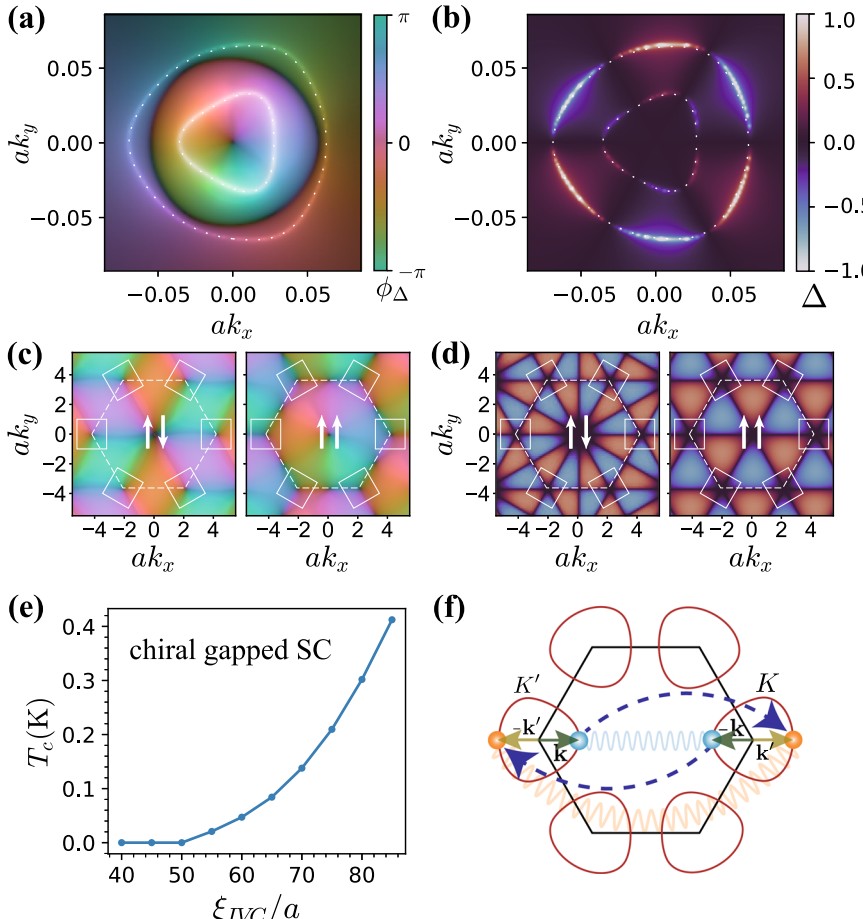

**Fig. 3 | Unconventional superconductivity from IVC fluctuations. a** Complex pair wave-function of the gapped chiral superconductor showing magnitude (intensity) and phase (hue). Coulomb repulsion favors a sign change between the interior and exterior Fermi surfaces (dotted white lines). **b** Real pair wave-function of nodal non-chiral superconductor with 6-fold oscillations around the annular Fermi surfaces. **c**, **d** Schematic depiction of the favored superconducting pairings extended from momenta patches around $K/K'$ points (white boxes) to the entire hexagonal BZ (dotted white lines), for spin-singlets (left) and triplets (right). **e** $T_c$ of the gapped chiral superconductor (e.g., $d + id$ spin-singlet) within

the self-consistent BCS calculations as a function of $\xi_{IVC}$. Calculations at density $n_e = -1.7 \times 10^{12}$ cm$^{-2}$, $u = 30$ meV, including the effect of Coulomb repulsion with screening $q_{TF} = \frac{e^2}{\epsilon} \chi_0$ and IVC fluctuations (Eq. (10)) of strength $g = n_e/\chi_0 \approx 6$ meV, where $\chi_0$ is the DOS at the Fermi energy. **f** Electron-scattering between valleys by IVC fluctuations (indicated by blue arrows), showing how an attractive interaction in the IVC channel is converted to a repulsive interaction between inter-valley Cooper pairs. Only one Fermi surface in each valley is shown. The Fermi surfaces are enlarged relative to the BZ for clarity.

valley-Zeeman field without losing significant interaction energy. This is again in accordance with our HF results, where the valley-polarized phase shows no local canting in the parameter regime where it is energetically favorable.

Experimentally, as the hole-density is further reduced towards neutrality there is another sequence of transitions, first to a spin-polarized and valley-unpolarized 'half-metal' (with zero spontaneous Hall resistance, $R_{xy} = 0$), subsequently to a second PIP phase, and finally to a spin and valley polarized 'quarter metal' (where $R_{xy} \neq 0$)[16]. While the Hall response of the intervening PIP phase is unknown, a reasonable candidate for this phase, which borders SC2, is a spin-polarized IVC metal, which HF calculations also show is competitive in this density region (see Supplementary Fig. 2). Starting with spin-polarized Fermi surfaces, the same interplay of kinetic energy benefit and interaction energy penalty can favor IVC over a valley-polarized state. Further reduction of hole-doping can suppress this kinetic energy gain, and tilt the energetic balance towards the observed spin-valley polarized 'quarter metal'.

## Hund's coupling

As alluded to previously, the inter-valley Hund's coupling plays a crucial role in determining the nature of iso-spin symmetry breaking. We

derive this term for an arbitrary translationally invariant interaction potential matrix $U_{\sigma\sigma'}(\mathbf{q})$ in the SM (See Supplementary Material), where $\sigma, \sigma'$ refer to $A_1/B_3$ sublattice indices within each unit cell. However, to illustrate the physical effect, we focus on a simple limit $U_{\sigma\sigma'}(\mathbf{q}) = U$, i.e., a local interaction $U$ that acts only within the unit cell and is independent of the sublattice index. In this limit, the Hund's coupling takes the form:

$$H_{\text{Hund's}} = -\frac{J_H}{A} \sum_{\mathbf{q}} \mathbf{s}_{+-}(\mathbf{q}) \cdot \mathbf{s}_{+-}^\dagger(\mathbf{q}) \tag{5}$$

where $\mathbf{s}_{+-}(\mathbf{q}) = \sum_{\mathbf{k}} \lambda_{\mathbf{q}}^{+-}(\mathbf{k}) \psi_{+,s,\mathbf{k}}^\dagger \mathbf{s}_{ss'} \psi_{-,s',\mathbf{k}+\mathbf{q}}$ is the inter-valley spin-density projected to the valence bands, and $J_H = U$. The Hund's coupling breaks the SU(2)$_+$ × SU(2)$_-$ symmetry down to the physical spin SU(2)$_s$ symmetry. While the short-range component of the Coulomb interaction is thus expected to give $J_H > 0$, other lattice-scale effects, such interactions between electrons and optical phonons, may also contribute: so we treat $J_H$ as a phenomenological parameter to be constrained by experiments.

For $J_H > 0$, the Hund's coupling term favors a triplet IVC, as $\mathbf{s}_{+-}$ is nothing but the triplet IVC order parameter $\mathbf{n}_T$. This can be understood by noting that a local repulsive interaction would disfavor excess

accumulation of charge that characterizes a CDW such as the singlet IVC. On the other hand, an attractive $U < 0$ favors the singlet IVC.

We note that $H_{\text{Hund's}}$ differs from another symmetry-allowed Hund's term $\tilde{H}_{\text{Hund's}} = -\frac{\tilde{J}_H}{A} \sum_{\mathbf{q}} \mathbf{s}_+(\mathbf{q}) \cdot \mathbf{s}_-(-\mathbf{q})$, where $\mathbf{s}_\tau$ is the spin-density in valley $\tau$. While $H_{\text{Hund's}}$ and $\tilde{H}_{\text{Hund's}}$ are related by a Fierz transformation at the lattice scale, after projection into the valence band they are not, giving rise to different physical effects. While a ferromagnetic $H_{\text{Hund's}}$ favors a triplet IVC state at the Hartree level as discussed above, $\tilde{H}_{\text{Hund's}}$ prefers either a spin-polarized or spin-valley locked state for $\tilde{J}_H > 0$ or $\tilde{J}_H < 0$ respectively. The difference between these two distinct Hund's terms is rooted in the opposite Berry curvature of the two valleys. Specifically, $\tilde{H}_{\text{Hund's}}$ contains only valley-diagonal form-factors $\lambda_{\mathbf{q}}^{\tau,\tau}(\mathbf{k})$, while the $H_{\text{Hund's}}$ derived microscopically from short-range $U$ has only valley-off-diagonal ones $\lambda_{\mathbf{q}}^{\tau,-\tau}(\mathbf{k})$. This is distinct from the SU(4) quantum Hall physics in monolayer graphene, where the Landau level wave-functions in both valleys have identical Berry curvature, in which case the two kinds of Hund's terms are related by Fierz identities[33]. However, for small momenta $\mathbf{k}$, the wavefunctions are nearly sublattice polarized, in which case the Berry curvature vanishes and the form factors become trivial, $\lambda^{\tau,-\tau}, \lambda^{\tau,\tau} \approx 1$. In this part of the BZ, the two types of Hund's terms *are* related by exchange symmetry. Therefore, at lower hole-doping, one might expect a lack of competition from kinetic energy and a small ferromagnetic $H_{\text{Hund's}}$ will tilt the balance in favor of spin-polarization. Indeed, a spin-polarized, valley-unpolarized 'half metal' phase is observed at hole dopings slightly lower than the spin-unpolarized PIP phase.

## IVC fluctuation-mediated superconductivity

**Superconducting instabilities.** Motivated by the likely presence of IVC order in the vicinity of superconductivity, we study superconducting instabilities mediated by near-critical fluctuations of the IVC order parameter. While the transition to the IVC state appears to be first order in the HF phase diagram of Fig. 2a, we find that the precise nature of this transition depends on details such as screening by the itinerant electrons; for example, small adjustments to $q_{\text{TF}}$ can render it continuous. Experimentally, there is no evidence of a first order phase transition (such as a negative compressibility spike) between the symmetric metal and the IVC metal, indicating that this transition is second order or weakly first order. To microscopically justify that IVC fluctuations are nearly gapless close to the transition, we compute the IVC correlation length $\xi_{\text{IVC}}$ within Hartree−Fock (see SM (See Supplementary Material) for details), and find that $\xi_{\text{IVC}}/a \approx 10^2$, i.e., $\xi_{\text{IVC}}$ becomes much larger than the microscopic lattice spacing $a$ near the transition. Therefore, we start in the symmetric metallic state with no long range IVC order, but with IVC correlations peaked at $\mathbf{q} = 0$. We assume that fluctuations of the IVC are described by phenomenological propagator of the form $g_{\mathbf{q}} = g/(\mathbf{q}^2 + \xi_{\text{IVC}}^{-2})$ at $\omega = 0$ (we provide an estimate of $g$ in the SM (See Supplementary Material)). In the spirit of spin-fermion models[34–36], we then integrate out the fluctuating IVC fields to obtain an effective inter-electron interaction. We first focus on the SU(2)$_+$ × SU(2)$_-$ symmetric case, where the effective interaction takes the form (Tr stands for tracing spin-indices):

$$H_{\text{IVC}}^{\text{eff}} = -\frac{1}{A} \sum_{\mathbf{q}} g_{\mathbf{q}} \, \text{Tr}\left[ n^{\text{IV}}(\mathbf{q})[n^{\text{IV}}(\mathbf{q})]^\dagger \right] \tag{6}$$

We use the above effective Hamiltonian as the pairing-interaction, in conjunction with the single-particle band structure projected to the valence band, to numerically solve a linearized BCS gap equation (See Supplementary Material for justification of projection, and further numerical details). We restrict attention to inter-valley pairing of the general form

$$F_{ss'}(\mathbf{k}) \equiv \langle \psi_{-,s,-\mathbf{k}} \psi_{+,s',\mathbf{k}} \rangle. \tag{7}$$

Intra-valley pairing occurs at finite center of mass momentum, and is expected to be energetically unfavorable.

Our numerical results are shown in Fig. 3. Remarkably, the leading superconducting instability is always towards a superconductor in which $F_{ss'}(\mathbf{k}) \approx -F_{ss'}(-\mathbf{k})$. It is tempting to call this 'odd-parity', but due to the valley degree of freedom the parity depends on whether the spin structure is singlet vs. triplet (recall that $\mathbf{k}$ is measured relative to the $K$ or $K'$ point). The precise pairing channel is sensitive to the correlation length $\xi_{\text{IVC}}$. For large $\xi_{\text{IVC}}$, pairing occurs first in the chiral $k_x \pm i k_y$ channels, leading to a fully gapped superconductor (at the mean-field level) with orbital angular momentum $L_z = \pm 1$ about the $K, K'$ points (Fig. 3a). The simplest extension of such an order parameter to the entire Brillouin Zone (BZ), consistent with fermionic anticommutation, is $d + id$ for spin-singlet, and $p + ip$ for spin-triplet (Fig. 3c)[37–41]. In contrast, a smaller $\xi_{\text{IVC}}$ leads to a non-chiral nodal superconductor with a gap-function $\sim k_y(3k_x^2 - k_y^2) = \text{Im}[(k_x + i k_y)^3]$ about the $K, K'$ points (Fig. 3b). We note that $C_3$ symmetry about the K point does not distinguish this nodal state from a trivial s-wave state ($L_z = 0$). Rather, such a gap function is odd under the combination of mirror $M_x$ and spinless time-reversal $\tilde{\mathcal{T}}$, leading to nodes at $k_y = 0$ and all $C_3$ related points: while an s-wave state is even under $M_x\tilde{\mathcal{T}}$ and non-nodal. The simplest extension of the nodal pairing function to the entire BZ involve a twelve-fold oscillation about the $\Gamma$ point (i-wave) for the spin-singlet, and a six-fold oscillation (f-wave) for the spin-triplet (Fig. 3d).

These results can be understood by analyzing the IVC fluctuation-mediated interaction in Eq. (6). Decoupling $H_{\text{IVC}}^{\text{eff}}$ in the Cooper channel,

$$\langle H_{\text{IVC}}^{\text{eff}} \rangle = \frac{1}{A} \sum_{\mathbf{k}, \mathbf{k}'} V_{\mathbf{k}\mathbf{k}'} \text{Tr}[F^\dagger(\mathbf{k})F(\mathbf{k}')] \tag{8}$$

where the effective interaction potential is $V_{\mathbf{k}\mathbf{k}'} = g_{\mathbf{q} = -\mathbf{k} - \mathbf{k}'} |\lambda_{\mathbf{q} = -\mathbf{k} - \mathbf{k}'}^{+-}(\mathbf{k})|^2$. When $\xi_{\text{IVC}}$ becomes large, $V_{\mathbf{k}\mathbf{k}'}$ is peaked at $\mathbf{q} = 0$. Thus, in contrast to the Coulomb interaction, IVC-induced scattering is strongest between Cooper pairs with *opposite* momenta $\mathbf{k} = -\mathbf{k}'$. An intuition for the resulting pairing channel is then gleaned from the $\mathbf{q} = 0$ limit of Eq. (8). Due to the SU(2)$_+$ × SU(2)$_-$ symmetry, spin-singlet superconductivity with $F(\mathbf{k}) = i s^y f_{\mathbf{k}}$ and unitary spin-triplet superconductivity with $F(\mathbf{k}) = (i s^y)(\hat{\mathbf{d}} \cdot \mathbf{s}) f_{\mathbf{k}}$ are degenerate. Inserting these ansatz into the $q \to 0$ limit,

$$\langle H_{\text{IVC}}^{\text{eff}} \rangle \approx \frac{2}{A} \sum_{\mathbf{k}} g_{\mathbf{o}} |\lambda_{\mathbf{q} = \mathbf{o}}^{+-}(\mathbf{k})|^2 f_{\mathbf{k}}^* f_{-\mathbf{k}} \tag{9}$$

Evidently, $\langle H_{\text{IVC}}^{\text{eff}} \rangle$ is minimized when $f_{\mathbf{k}}^* = -f_{-\mathbf{k}}$, corresponding to unconventional pairing, as found in our numerical calculations. This result is reminiscent of Cooper-pairing due to spin fluctuations in $C_4$ symmetric systems, such as high-$T_c$ cuprates, where a repulsive interaction leads to sign-change of the pairing order parameter between points on the Fermi surface connected by the wavevector where the spin flutuations are strongest, resulting in a d-wave superconductivity[42]. In $C_3$ symmetric RTG, inter-valley scattering by IVC fluctuations mediates an analogous repulsive interaction between inter-valley Cooper pairs[43], and leads to sign-change in $f_{\mathbf{k}}$ across the Fermi surface within each valley (see Fig. 3f for a schematic depiction).

Next, we turn to the $\xi_{\text{IVC}}$-induced transition between chiral gapped and non-chiral nodal superconductivity. When $\xi_{\text{IVC}}$ is large, the effective interaction strength $g_{\mathbf{q}}$ becomes increasingly singular at small $|\mathbf{q}|$. In this regime, the fully gapped $f_{\mathbf{k}} \sim k_x \pm i k_y$ is most energetically favorable, since it has a uniform magnitude of the gap on the Fermi surface, and gains the most from the singular part of the interaction. Further, the pairing amplitude is typically stronger on the inner Fermi surface (see Fig. 3a), which hosts a larger density of states. In contrast, when $\xi_{\text{IVC}}$ is small, $g_{\mathbf{q} = \mathbf{o}} \approx g \, \xi_{\text{IVC}}^2$ and $V_{\mathbf{k}\mathbf{k}'}$ is determined by the inter-

valley form factor $|\lambda^{+-}_{\mathbf{q}=\mathbf{0}}(\mathbf{k})|^2$. The form-factor has a six-fold oscillating structure across the Fermi surface, which induces an corresponding oscillating structure in $f_{\mathbf{k}}$, leading to the nodal superconductor observed numerically. In this case, pairing is much stronger on the outer Fermi surface which is at larger momenta, as opposed to the inner Fermi surface where the layer polarization term dominates and $|\lambda^{+-}_{\mathbf{q}=\mathbf{0}}(\mathbf{k})|^2$ is approximately constant (see Fig. 3b). These considerations explain the $\xi_{\mathrm{IVC}}$-induced transition between preferred superconducting channels.

Figure 3e shows the mean-field $T_c$ as a function of the correlation length $\xi_{\mathrm{IVC}}$ for the chiral superconducting state, including the effect of long-range Coulomb repulsion (See Supplementary Material for further details of this calculation). We find that $T_c$ is a strongly increasing function of $\xi_{\mathrm{IVC}}$, and as a result $T_c$ is appreciable only in the regime where the fully-gapped chiral state dominates. We therefore expect that this state, which is $d + id$ ($p + ip$) for spin-singlet (spin-triplet), is the one realized in the experiments. We note that in this calculation, we have ignored the frequency dependence of the interaction, and the damping of the electrons by bosonic IVC fluctuations. Both effects are known to become important close to the critical point, and we defer a detailed study of these effects to future work[44].

**Effect of Hund's coupling.** The inter-valley Hund's coupling splits the degeneracy between spin-singlet and spin-triplet superconductors, by amplifying SDW IVC fluctuations over CDW IVC fluctuations or vice versa, depending on the sign of $J_H$. To see this, we use the Fierz identity $2\delta_{\alpha\nu}\delta_{\beta\mu} = \mathbf{s}_{\alpha\beta} \cdot \mathbf{s}_{\mu\nu} + \delta_{\alpha\beta}\delta_{\mu\nu}$ to decompose the effective Hamiltonian for IVC fluctuations into of spin-singlet and spin-triplet IVC channels:

$$H^{\mathrm{eff}}_{\mathrm{IVC}} = -\frac{1}{2A}\sum_{\mathbf{q}}\left( g^{\mathrm{T}}_{\mathbf{q}}\,\mathbf{n}^{\mathrm{IV}}_{\mathrm{T}}(\mathbf{q})\cdot[\mathbf{n}^{\mathrm{IV}}_{\mathrm{T}}(\mathbf{q})]^{\dagger} + g^{\mathrm{S}}_{\mathbf{q}}\,n^{\mathrm{IV}}_{\mathrm{S}}(\mathbf{q})[n^{\mathrm{IV}}_{\mathrm{S}}(\mathbf{q})]^{\dagger}\right) \quad (10)$$

In the $\mathrm{SU}(2)_+ \times \mathrm{SU}(2)_-$ symmetric limit, the susceptibilities $g^{\mathrm{S}}_{\mathbf{q}} = g^{\mathrm{T}}_{\mathbf{q}}(=g_{\mathbf{q}})$ for the singlet and triplet IVC states are identical. However, including Hund's coupling breaks this symmetry and amplifies one susceptibility at the expense of the other, so more generally $g^{\mathrm{S}}_{\mathbf{q}} \neq g^{\mathrm{T}}_{\mathbf{q}}$, and we have:

$$\langle H^{\mathrm{eff}}_{\mathrm{IVC}}\rangle \approx \begin{cases} \frac{1}{A}\sum_{\mathbf{k}}(3g^{\mathrm{T}}_{\mathbf{0}} - g^{\mathrm{S}}_{\mathbf{0}})|\lambda^{+-}_{\mathbf{q}=\mathbf{0}}(\mathbf{k})|^2 f^*_{\mathbf{k}} f_{-\mathbf{k}}, \text{singlet SC} \\ \frac{1}{A}\sum_{\mathbf{k}}(g^{\mathrm{T}}_{\mathbf{0}} + g^{\mathrm{S}}_{\mathbf{0}})|\lambda^{+-}_{\mathbf{q}=\mathbf{0}}(\mathbf{k})|^2 f^*_{\mathbf{k}} f_{-\mathbf{k}}, \text{triplet SC} \end{cases} \quad (11)$$

From Eq. (11), we see that when triplet-IVC fluctuations are stronger, i.e., $g^{\mathrm{T}}_{\mathbf{q}} > g^{\mathrm{S}}_{\mathbf{q}}$, a spin-singlet superconductor becomes energetically favorable. Since a triplet IVC state is preferred by ferromagnetic Hund's coupling arising from short-range repulsive interactions ($J_H > 0$), this leads to the surprising conclusion that such a Hund's coupling also prefers a spin-singlet superconductor.

Intuitively, this happens because ferromagnetic Hund's coupling promotes antiferromagnetic fluctuations that couple antipodal points on the Fermi surface, promoting singlet superconductivity with an order parameter that changes its phase between these points, in analogy to the cuprates[42] and magic angle twisted bilayer graphene[43,45]. In contrast, an antiferromagnetic Hund's term amplifies singlet-IVC fluctuations with $g^{\mathrm{S}}_{\mathbf{q}} > g^{\mathrm{T}}_{\mathbf{q}}$, and therefore leads to a spin-triplet p/f wave perturbatively away from the fully symmetric point. When it significantly enhances singlet-IVC fluctuations, the effective interaction $V_{\mathbf{k},\mathbf{k}'}$ turns attractive and a spin-singlet fully-gapped s-wave superconductor becomes the most favored pairing channel.

If we assume that the sign of the Hund's term does not change across the doping range studied in the experiment, we expect it to be ferromagnetic since it prefers spin-polarization at low doping. This

leads to the interesting prediction that SC1 is a spin-singlet chiral $d + id$ superconductor. This conclusion is consistent with fact that SC1 obeys the Pauli limit[9]. Of course, as discussed previously, such a ferromagnetic Hund's term may also drive a transition to a spin-polarized IVC state, as possibly happens at lower doping. In this case, IVC fluctuations favor a spin-polarized (triplet) state, which we consider a candidate for SC2.

**Effect of Coulomb repulsion.** Finally, we comment on the effect of Coulomb interactions in our numerical solutions of the BCS gap equation. Some intuition can be gained by analyzing $H_C$ at a mean-field level, by decoupling the Coulomb interaction in the Cooper channel:

$$\langle H_C\rangle = \frac{1}{A}\sum_{\mathbf{k},\mathbf{k}'} V^c_{\mathbf{k},\mathbf{k}'}\,\mathrm{Tr}[F^{\dagger}(\mathbf{k})F(\mathbf{k}')] \quad (12)$$

where $V^c_{\mathbf{k},\mathbf{k}'} = |\lambda^{++}_{\mathbf{q}=\mathbf{k}'-\mathbf{k}}(\mathbf{k})|^2 V_C(\mathbf{q}=\mathbf{k}'-\mathbf{k})$ is the effective repulsive potential. The repulsion from Eq. (12) with static RPA screening was included in the BCS calculations for $T_c$ shown in Fig. 3(c).

Noting that $V_C(\mathbf{q})$ and $|\lambda^{++}_{\mathbf{q}}(\mathbf{k})|^2$ are positive and peaked at $\mathbf{q} = 0$, the $\mathbf{k} \to \mathbf{k}'$ limit gives a large contribution to Eq. (12). Since $\mathrm{Tr}[F^{\dagger}(\mathbf{k})F(\mathbf{k})]$ is always positive semi-definite, this leads to the expected conclusion that a repulsive Coulomb interaction disfavors superconductivity in all channels. However, for annular Fermi-surfaces, the superconductor can reduce the Coulomb penalty by flipping the sign of the pairing between the outer and inner Fermi surfaces, while leaving the pairing symmetry unchanged. This leads to an attractive contribution to Eq. (12) for wavevectors $\mathbf{q}$ which connect the inner and outer Fermi surfaces. This sign change is indeed found in the solution to the linearized BCS equations shown in Fig. 3a. Furthermore, we find that the gapped chiral superconductor is quite robust to Coulomb interactions, indicating that strong near-critical IVC fluctuations can overcome repulsion between electrons and lead to Cooper-pairing. In contrast, the Coulomb interaction destabilizes the *weaker* pairing in the nodal superconductor in favor of a metallic phase.

## Discussion

In this paper, we showed that IVC metallic phases, with and without net spin-polarization, are promising candidates for the symmetry broken phases adjacent to the SC2 and SC1 superconductors respectively. Fluctuations in the IVC order parameter can provide the pairing glue for superconductivity in RTG, with $T_c$ comparable to experiments. IVC fluctuations naturally favor gapped chiral superconductivity or non-chiral nodal superconductivity, depending on the correlation length $\xi_{\mathrm{IVC}}$. In the $\mathrm{SU}(2)_+ \times \mathrm{SU}(2)_-$-symmetric model, the spin-singlet and triplet channels are degenerate. The short-range Hund's coupling which breaks this symmetry then favors either (1) an IVC corresponding to a spin-singlet CDW, and triplet superconductivity or (2) an IVC corresponding to a spin-triplet SDW, and singlet superconductivity. The latter superconductor breaks only $\mathrm{U}(1)_c$, and has a finite temperature BKT transition, and is Pauli limited, consistent the experimental observations for SC1.

On the other hand, fully spin-polarized IVC fluctuations at lower hole-densities can lead to a spin-polarized chiral or nodal superconductor, consistent with the Pauli limit violation observed for SC2. We note that such a superconductor has an order parameter manifold of $\mathrm{SO}(3)$[20–22], which would not have a finite temperature BKT transition in absence of a Zeeman field. However, if the magnetic correlation length is large enough, we expect apparent superconducting behavior for low enough temperatures and finite-size systems.

### Experimental probes

To experimentally verify the IVC metal in RTG, we note that it is either a CDW, or a SDW with a small CDW component. Thus spin-polarized scanning tunneling microscopy (STM)[46,47] is the probe of choice, as it

can directly access the spin and charge density distribution at the lattice scale. However, since symmetry considerations do imply that the SDW will induce a weak CDW, a good first step is spin-unpolarized STM, where a tripled unit cell should be observable in the site-resolved LDOS.

Our theory predicts that the superconducting phases are unconventional in nature, in the sense that the average of the order parameter over the Fermi surface vanished. Such an order parameter is expected to be sensitive to small amounts of non-magnetic disorder[48,49]. The chiral phase should produce spontaneous edge currents[50], observable in scanning nano-SQUID experiments. However, we carefully note that a chiral superconductor obtained from a parent metal with an annular Fermi surface is topologically trivial. To see this, we consider the BdG mean-field spectrum of the superconductor, where we first tune the chemical potential to empty all the bands, and subsequently tune the superconducting gap to zero. The chiral order parameter is gapless only at $K/K'$ points, which never touch the annular Fermi surface as $\mu$ is tuned. Thus, the bulk BdG gap never closes during this process, implying that the chiral superconductor is smoothly connected to the topologically trivial vacuum. Hence, we do not expect quantized edge modes, though the $\mathcal{T}$-breaking may still manifest in a bulk magnetization observable as edge currents. Finally, current-noise spectroscopy using quantum impurity defects[51] can efficiently distinguish between nodal and fully gapped chiral superconductors[52,53].

### Alternative routes to superconductivity
Alternative mechanisms of superconductivity are possible, and deserve further investigation. Ref. 54 studies inter-electron attraction mediated by acoustic phonons as a possible pairing mechanism, and finds s-wave spin singlet/f-wave spin triplet superconductors to be favored. However, acoustic phonons do not choose between a singlet and a triplet superconductor, as the phonon-mediated interactions are fully $SU(2)_+ \times SU(2)_-$ symmetric (optical phonons do not preserve this symmetry, but coupling of low-energy electrons to optical phonons is very weak in RTG under strong displacement fields[55]). Suppose we could characterize the phase diagram by a single Hund's coupling $J_H$. Then, the presence of spin-polarized, valley-unpolarized phases in the phase diagram indicates that $J_H$ is ferromagnetic. In such a scenario, a pairing mechanism based solely on acoustic phonons would predict a spin-triplet superconductor, in contradiction with the experimental observation for SC1. Our proposed scenario can explain both the presence of spin-polarized phases and spin-singlet superconductivity within a single, consistent picture. Further, we note that the same acoustic phonons would act as an external bath for electrons, and lead to a strong linear in T resistivity in the metallic state above the Bloch-Grüneisen temperature, which has not been observed in RTG[9]. While isospin fluctuations can also potentially increase the resistance above $T_c$, these fluctuations microscopically originate from the collective behavior of the electrons themselves. Therefore, these result in electron-electron scattering that strongly affects single-particle lifetimes, but does not degrade the net momentum (in absence of umklapp scattering[56]). Thus, collective isospin fluctuations can only contribute to d.c. transport in the presence of disorder. We leave this interesting problem to future work.

On a different note, a two-dimensional annular Fermi surface allows for a Kohn–Luttinger mechanism for pairing[57–61]. Similarly to the mechanism explored in this work, in the Kohn–Luttinger mechanism the pairing is driven by electronic fluctuations. However, no particular soft collective mode is assumed (i.e., the system is not assumed to be close to a continuous transition). Instead, all the particle-hole fluctuation channels contribute on the same footing. For RTG, this mechanism was recently found to lead to a chiral state[62], similar to the state predicted in this work in the vicinity of the critical point.

### Outlook
Our study provides a starting point for further theoretical and experimental investigation of correlation effects and superconductivity in RTG in particular, and in *non-moiré* few-layered graphene more generally. It also shows that, somewhat contrary to usual belief, spin-singlet superconductors can be favored by ferromagnetic Hund's coupling when additional (valley) degrees of freedom are relevant. While our phenomenological treatment of coupling between electrons and soft-modes only allows us obtain an estimate of the superconducting critical temperature, our work motivates numerical explorations to determine $T_c$ accurately as a function of carrier density and electric field in RTG. Understanding the relevance of RTG physics to moiré graphene platforms, which also feature strong iso-spin fluctuations in topological flat bands[30,63–65], or to surface superconductivity in rhombohedral graphite[66,67] is left for future work.

### Note added
Recently, we became aware of another study of isospin fluctuation-mediated superconductivity in RTG[68]. Since this paper was submitted, several more studies of unconventional superconductivity in RTG have appeared[69–71].

### Data availability
All data generated or analyzed during this study are included in this published article (and its Supplementary Information files).

### Code availability
The codes used to generate the plots are available from the corresponding author on reasonable request.

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

## Acknowledgements

We thank A. Black-Schaffer, A. Vishwanath, S. Whitsitt, Y. You and A. F. Young for helpful discussions. S.C. and M.Z. were supported by the ARO through the MURI program (grant number W911NF-17-1-0323). S.C. also acknowledges support from the U.S. DOE, Office of Science, Office of Advanced Scientific Computing Research, under the Accelerated Research in Quantum Computing (ARQC) program via N.Y. Yao. T.W. and M.Z. were supported by the U.S. DOE, Office of Science, Office of Basic Energy Sciences, Materials Sciences and Engineering Division, under Contract No. DE-AC02-05CH11231, within the van der Waals Heterostructures Program (KCWF16). E.B. was supported by the European Research Council (ERC) under grant HQMAT (Grant Agreement No. 817799), by the Israel-USA Binational Science Foundation (BSF), and by a Research grant from Irving and Cherna Moskowitz.

## Author contributions

All authors contributed extensively to all aspects of this work.

## Competing interests

The authors declare no competing interests.
