## [Peer Review File · Nature Communications]

Reviewers' Comments:

Reviewer #1:

Remarks to the Author:

This review concerns "Inter-valley coherent order and isospin fluctuation mediated superconductivity in rhombohedral trilayer graphene" (NCOMMS-21-41949-T) by Chatterjee and coworkers.

The work is motivated by recent experiments in which superconductivity was observed in ABC-stacked trilayer graphene in certain regions of the doping - displacement field parameter space. Specifically, superconductivity was found to appear at the phase boundaries of two broken symmetry phases, thus raising interesting and important questions about the natures of superconducting phases and their relation to the proximal broken symmetry phases.

The present work (NCOMMS-21-41949-T) seeks to address these questions from a theoretical point of view, through Hartree-Fock calculations in combination with analysis of constraints imposed by the experimental observations (e.g., sensitivity to in-plane magnetic field). The Hartree-Fock results support the assignment of the symmetry-broken metallic phases as arising from spontaneous intervalley coherence (which was deduced from the lack of observed spin or valley polarization in the experiment).

Physical signatures of the spin-singlet and spin-triplet intervalley coherent states (CDW and CDW/SDW order) are discussed.

The authors also provide an illuminating physical argument to explain why intervalley coherence is energetically more favorable than valley polarization (which would be an alternative candidate for the broken symmetry).

They also demonstrated that fluctuations of the intervalley coherence order parameter can indeed mediate superconductivity, with a mean-field transition temperature of similar order to that observed in experiments, and discussed how Hund's coupling affects the specific type of IVC order and superconducting pairing expected.

The study is timely, thorough, and clearly presented.

An extensive supplement is provided with all technical details needed to reproduce the work.

The analysis and calculations also appear to be of very high quality, and shed important light onto the recently discovered phases in ABC graphene.

For these reasons, I am happy to recommend publication in Nature Communications, once the small comments/questions below are addressed.

1) In Sec. IV, which discusses superconductivity mediated by fluctuations of the intervalley coherence order parameter, analysis is carried out within a projection onto the valence band. Regarding justification of the projection, how large are the relevant interaction energies compared with the displacement field induced gap?

2) There are a few typos that should be fixed. For example, above Eq. (3), "Defining" should be "We define," and on page 4: "... and spin-triplet SDW ..." should be "... or the spin-triplet SDW ...". One more close read should be enough to catch the rest.

Reviewer #2:

None

Reviewer #3:

Remarks to the Author:

The authors present a theory for explaining the superconductivity in ABC TLG. The work is timely and valuable, but many questions need to be clarified before I could consider a suggestion.

1. Superconductivity in R-graphite has been theoretically studied by quite a few groups before Young's experiments.

2. Can the author show the E-field dependence and estimate the T_c ? These should be compared with the experimental data.
3. Are there other possible IVC phases and how were they be excluded?
4. Can the authors elucidate the e-p couplings and explain why BCS superconductivity should be excluded?

Point-by-point response to referee A:

The study is timely, thorough, and clearly presented. An extensive supplement is provided with all technical details needed to reproduce the work. The analysis and calculations also appear to be of very high quality, and shed important light onto the recently discovered phases in ABC graphene.

We thank the referee for the encouraging words and positive evaluation of our work, and note that they concur with Ref C on the study being timely and valuable to the community.

In Sec. IV, which discusses superconductivity mediated by fluctuations of the intervalley coherence order parameter, analysis is carried out within a projection onto the valence band. Regarding justification of the projection, how large are the relevant interaction energies compared with the displacement field induced gap?

This is an excellent question. Near the K point, the bands are quite flat and the displacement-field induced gap between the valence band and the conduction band is around 64 meV at $u = 30$ meV. The typical interaction scale, for unscreened Coulomb interaction, is given by $E_C = e^2/4\pi\epsilon\epsilon_0\langle r \rangle$, where $\langle r \rangle \approx 1/\sqrt{n_h}$ is the average separation between the charge carriers. For the experimentally relevant carrier-density $n_h \sim 10^{12}$ cm⁻² and hBN dielectric constant $\epsilon = 4.4$, we find that $E_C \approx 33$ meV, which is smaller than the band-gap, albeit not by an order of magnitude. Dual-gate screening reduces this estimate to $E_C \approx 28$ meV, and screening due to itinerant hole-carriers in the sample itself will further decrease E_C . In addition, the matrix elements coming from imperfect wavefunction overlaps between the valence and conduction bands would act to further suppress interband scattering. Therefore, it is reasonable to project the interaction onto the valence band to analyze fluctuation-mediated superconductivity. We have added the above justification of this projection in the Appendix F2, where we derive the gap equation.

For the sake of completeness, we also note that the *active* valence band and the next remote valence band is 320 meV, which is much larger than the interaction scale - so neglecting these bands are well-justified.

There are a few typos that should be fixed. For example, above Eq. (3), "Defining" should be "We define," and on page 4: "... and spin-triplet SDW ..." should be "... or the spin-triplet SDW ..." One more close read should be enough to catch the rest.

We thank the referee for pointing out the typos, and have fixed these in the revised version

Point-by-point response to referee B:

While this study is timely, I am not convinced that the manuscript should be published in a high-impact journal like Nature Communications. The main concerns are as follows.

We thank the referee for their appreciative comment about the work being timely, and for raising several important questions. We address all the concerns point by point below.

As far as I understand, the idea of IVC fluctuation as a possible pairing glue is at best a wishful thinking at this stage.

We respectfully but firmly disagree with the referee that the idea of IVC fluctuation as a pairing glue is wishful thinking. In the experiments by Zhou *et al*, there is definitive evidence of isospin symmetry-breaking in the metallic state. Given that superconductivity is only seen adjacent to symmetry-breaking, order parameter fluctuations are natural to consider as a pairing glue for superconductivity. Our reason for considering IVC fluctuations specifically is not solely Hartree-Fock numerics, but strong experimental evidence that the adjacent symmetry-broken state is an IVC. To see this, we note that this symmetry-broken state has been shown to be neither spin-polarized (the phase boundary with the fully symmetric state does not shift with an in-plane magnetic field), nor valley-polarized (no Hall conductivity). Further, the quantum oscillation data indicates majority and minority carrier Fermi surfaces, which are present in an IVC state (see Fig. 1d). Therefore, there are strong constraints in the experimental phase diagram that point towards an IVC phase lying adjacent to the superconducting phase.

For example, this mechanism requires the transition from the symmetry-preserving state to the symmetry-breaking state to be continuous, which, however, needs fine-tuning of model parameters as the authors admit.

We thank the referee for this important comment. To address this, we note that for superconductivity to be mediated by IVC fluctuations, it is not strictly necessary for the phase transition between the normal metal and the IVC phase to be second order. As long as the IVC correlation length is reasonably large ($\xi_{IVC}/a \gtrsim 60$), we find that the T_c for the superconductor thus obtained is consistent with experiments. Thus, a weakly first order phase transition will also suffice.

In order to demonstrate that the correlation length is sufficiently large, we undertook a computation of ξ_{IVC} using Hartree-Fock. Within a single-mode approximation, we can compute the correlation length ξ_{IVC} , the results are shown in the figure below. We find that the IVC fluctuations are singular enough near the transition to justify the mechanism we propose.

We would also like to point out that experimentally, there is no evidence of the phase transition into the isospin symmetry-broken phase being first order. For instance, there is no negative compressibility spike, nor is there any hysteresis (which is present in other parts of the phase diagram) that would indicate that a first order transition.

In the revised version, we have added a new section (Appendix F4) which details the computation of the IVC correlation length, and the following discussion to the main text:

Experimentally, there is no evidence of a first order phase transition (such as a negative compressibility spike) between the symmetric metal and the IVC metal, indicating that this transition is second order or weakly first order. To microscopically justify that IVC fluctuations are nearly gapless close to the transition, we compute the IVC correlation length ξ_{IVC} within Hartree Fock (see SM [?] for details), and find that $\xi_{IVC}/a \approx 10^2$, i.e., ξ_{IVC} becomes much larger than the microscopic lattice spacing a near the transition.

In principle, there are many other fluctuations, such as spin fluctuation and valley polarization fluctuation.

While fluctuations in other isospin sectors are certainly possible, we require the fluctuation to be *near-critical* to provide the pairing glue by overcoming the Coulomb repulsion between the electrons. As discussed in the response to the previous comment, there is clear experimental evidence that the adjacent PIP (partially isospin polarized) metallic phase is neither spin nor valley polarized. Both our microscopic computation and analytical arguments point towards an IVC phase, which is consistent with the experimental observations. Therefore, there is no reason to expect spin or valley fluctuations to be significantly enhanced relative to IVC fluctuations.

IVC fluctuation as a possible pairing glue has been suggested previously, for example, in PHYSICAL REVIEW X 8, 031089 (2018) and npj Quantum Materials 4, 16 (2019)

We completely agree with the referee that IVC fluctuation as a possible pairing glue has been suggested in the context of twisted bilayer graphene (TGB) (we have cited both of these papers). However, our work differs from these in several aspects, which we elaborate on below.

In our view, the physics in RTG is much more strongly experimentally constrained than in TBG. Further, the quoted papers do not present any microscopic motivation for preferring IVC fluctuations. In our work, we provide both phenomenological and microscopic reasons for the possibility of a proximate IVC phase. Further, from a microscopic derivation of the inter-valley Hund's coupling, we demonstrate that it directly favors an IVC state among different isospin symmetry-broken states. Finally, our added computation of the IVC correlation length strengthens the case for nearly critical IVC fluctuations near the transition. Taken together, we contend that the case for IVC fluctuation mediated superconductivity is much stronger in RTG.

Let us now elaborate on the differences between our work and previous literature on IVC fluctuation mediated superconductivity.

- The paper by Po *et al* consider Goldstone modes of the IVC fluctuations. Their starting point, thus, is the symmetry-broken IVC state in TBG, which in turn leads to superconductivity mediated via its gapless fluctuations. This is in stark contrast to our work, where we consider gapped 'paramagnon'-like fluctuations of the IVC order parameter as the pairing glue. Further, we carefully take into consideration the form-factors arising band-projection, which have been shown to have significant effect on the physics on Van der Waals materials including TBG (see, for example, Bultinck *et al*, PRX 2019).
- The paper by You *et al* considers superconductivity mediated by IVC fluctuations in TBG, but their work differs from ours in many significant aspects. First, the driving force of IVC in their work is Fermi surface nesting. However, in RTG, there is no significant Fermi surface nesting, and rather an IVC state is energetically close to spin or valley-polarized phases if the inter-valley Hund's coupling is neglected.

Once this Hund’s coupling is taken into account, the IVC phase is preferred over other isospin symmetry broken phases. Secondly, the form-factors arising from band-projection were not considered in You *et al*, to the best of our understanding. We showed that taking form factors into account can lead to either to chiral gapped superconductivity or non-chiral nodal superconductivity, depending on the strength of the fluctuations. Finally and most importantly, the superconductor obtained in TBG from IVC fluctuations has topologically protected edge-modes. Because of the annular Fermi surface of the parent metallic state in RTG, there would not be any topologically protected edge-modes in the chiral superconductor, as we have argued in the discussion section.

To conclude, we are not claiming that superconductivity mediated by IVC fluctuations is itself a new idea. Indeed, superconductivity from low-energy fluctuations of a bosonic order parameter is well-known in the literature (e.g., spin-fluctuations in the cuprates). Rather, we have used microscopic and experimental constraints to nail down IVC fluctuations as a likely mechanism for superconductivity in RTG. We have also shown, via explicit computation of the IVC correlation length and numerical solution to the superconducting gap equation, that a realistic T_c may be achieved via this mechanism (this is again absent in the quoted papers). Finally, with a single consistent assumption of ferromagnetic inter-valley Hund’s coupling, our mechanism can naturally explain the spin-singlet nature of superconductivity lying close to a spin-polarized half-metal phase. This provides a plausible resolution to an important experimental puzzle in RTG, that, for example, cannot be explained by phonon-mediated superconductivity.

The authors compare the energy of different states in fig. 2(a) for a fixed layer potential difference. Could the authors present a phase diagram as a function of the layer potential difference and the electron density? To what extent can the experimental phase diagram be reproduced theoretically?

We thank the author for this question. We have taken their suggestion seriously, and explored the phase diagram as a function of electric field and doping using Hartree-Fock numerics. In the figure above, we present the Hartree-Fock phase diagram on the left (we consider only the fully symmetric and IVC phases) and the actual experimental phase diagram on the right. Despite of a systematic shift of the phase boundary toward lower doping (this could be attributed to neglecting the Hund’s term which prefers an IVC phase), the broad qualitative features of the phase diagram agree well with the experiments. First, there is no isospin symmetry-breaking at zero displacement field. Secondly, the phase boundary is nearly a straight line at large displacement field, and flattens out at smaller displacement fields.

The authors use a screened Coulomb interaction within the TF approximation based on the noninteracting density of states. However, the DOS changes once interaction effects are taken into account. For example, the DOS is different for the symmetry-preserving state and an isospin polarized state. Can the authors make some comments on this approximation? If unscreened Coulomb interaction is used, how different are the results?

This approximation is well justified close to the phase transition point since the Fermi surface, and thus DOS, changes only slightly as the transition is either weakly first order or second order in the parameter regime of interest. In the opposite limit, when the system is sufficiently far from the transition, and energy difference between the symmetric state and the iso-spin polarized states becomes so large that we only need to worry about the competition among different iso-spin polarized states. In this regime, the two energetically most favorable states, i.e. the spin polarized state and the intervalley coherent state have almost identical Fermi surfaces - thus choosing a different DOS in Hartree-Fock numerics only changes their energy together.

If we use unscreened Coulomb interaction, the state predicted by Hartree-Fock numerics, especially in the low doping regime, deviates significantly from the quantum oscillation experiment. In particular, unscreened Coulomb interaction never favors a Fermiology with 3-pockets or an annular Fermi surface (per valley per spin), both of which are observed close to charge neutrality in quantum oscillation experiments. Thus, we need to account carefully for screening of Coulomb interaction so that we can reproduce the observed Fermiology of RTG within Hartree-Fock numerics.

For Eq. (B1) that presents the RPA approximation, can the authors check the denominator? Is it $1-\chi*V$ or $1+\chi*V$?

We thank the referee for pointing this out. We previously used a slightly different convention, and defined $\chi_{\rho\rho}$ with an additional minus sign. We have reverted to the more conventional definition of $\chi_{\rho\rho}$ in the revised version. The change does not alter any of our conclusions.

Point-by-point response to referee C:

The authors present a theory for explaining the superconductivity in ABC TLG. The work is timely and valuable, but many questions need to be clarified before I could consider a suggestion.

We thank the referee for the appreciative comments and their list of questions which have helped us improve the clarity of the manuscript.

Superconductivity in R-graphite has been theoretically studied by quite a few groups before Young's experiments.

We thank the referee for pointing out previous theoretical studies of superconductivity in R-graphite. We have added references to previous works that we could find, and would certainly be happy to include additional references that we may have missed. We would also like to highlight that there are some important difference between our study and the previous studies.

First, to the best of our knowledge, the previous studies predicted superconductivity *in absence of a perpendicular electric field*. However, the experimental phase diagram of Zhou *et al* shows that without the electric field, the correlation effects are absent and there is no superconductivity in RTG in the accessible temperature range. This implies that the electric field is required to further enhance the density of states (beyond the increased density of states due to the cubic band touching) and observe correlated physics. The presence of the electric field modifies the wavefunctions in a crucial way (e.g., makes them strongly sublattice polarized near the K/K' points), as we discuss in our paper.

Second, the pairing symmetry predicted in previous works is predominantly s-wave, while in our work we find a clear case for unconventional (chiral) superconductivity.

Third, some previous works (e.g., Kopnin *et al*, PRB 87, 140503(R) (2013)) predict that superconductivity is in the ultra strong-coupling regime, with T_c comparable to the Fermi energy ε_F . In RTG, the experimentally observed ratio T_c/ε_F is quite small, consistent with the fluctuation-mediated scenario we have proposed.

Finally, in the ultra strong-coupling regime, it is unclear if the metallic state proximate to the superconductor can be a simple Fermi liquid. However, in RTG - quantum oscillations show definitive evidence of Fermi pockets with degeneracies corresponding to spin and/or valley degrees of freedom. Based on this, it appears that RTG may lie away from the ultra-strong coupling regime.

Can the author show the E-field dependence and estimate the T_c ? These should be compared with the experimental data.

We thank the author for this important question. We have now calculated the phase diagram as a function of both electric field and doping. In the figure above, we present the Hartree-Fock phase diagram on the left and the actual experiment phase diagram on the right. Despite of a systematic shift of the phase boundary toward lower doping due to the lack of Hund's term, the broad qualitative features of the phase diagram agree well with the experiments. In particular, the phase boundary is a nearly straight line at large displacement field, and there is no isospin symmetry-breaking at zero displacement field.

We have added the left panel as Fig. 2(b), accompanied by the following discussion, to the main text:

The resulting phase diagram as a function of hole-doping and displacement field is presented in Fig. 2(b), and a line cut at a fixed displacement field is shown in Fig 2(a). ... Nevertheless, we note that the broad features of our phase diagram (Fig. 2(b)), such as interaction-induced symmetry breaking at large displacement fields, and the phase boundary between the spin-unpolarized IVC metal and the fully symmetric metal, are consistent with experiments.

The interaction Hamiltonian that we consider for the purpose of investigating superconductivity is phenomenological, and the coupling g between the order parameter fluctuations and the electrons is not self-consistently determined in our work. Thus, while we are able to obtain an estimate of T_c and show that it is in the same ballpark as experimental observations, our approach cannot precisely predict T_c . We believe that accurately solving for T_c requires different analytical/numerical tools altogether, and we do not attempt to do so in this paper.

Are there other possible IVC phases and how were they be excluded?

This is an interesting question, and we thank the referee for asking it. As we discuss in the text, the most general IVC order parameter, projected to the valence band, is a 2×2 matrix in spin-space:

$$n_{ss'}^{IV}(\mathbf{q}) = \sum_{\mathbf{k}} \lambda_{\mathbf{q}}^{+-}(\mathbf{k}) \psi_{+,s,\mathbf{k}}^\dagger \psi_{-,s',\mathbf{k}+\mathbf{q}} \quad (1)$$

In absence of the inter-valley Hund's coupling, spin-singlet and spin-triplet IVC order parameters can mix due to independent spin-rotation symmetry in each valley. However, since the Hund's coupling plays an important role in the physics of RTG, we will only consider the global $SU(2)$ spin-rotation, under which $n_{ss'}^{IV}(\mathbf{q})$ transforms either as a spin-singlet ($n_{ss'}^{IV} \propto \delta_{ss'}$) or a spin-triplet ($n_{ss'}^{IV} \propto (\hat{\mathbf{n}} \cdot \mathbf{s})_{ss'}$ for an arbitrary unit-vector $\hat{\mathbf{n}}$).

With the above restriction on the IVC order parameter, let us first consider order at momentum $\mathbf{K} - \mathbf{K}'$,

corresponding to non-zero expectation value of $n_{ss'}^{IV}(\mathbf{q} = 0)$. These correspond to the 3-sublattice charge-density wave (CDW) for the spin-singlet, and the collinear 3-sublattice spin-density wave (SDW) state, as discussed in our paper. As long as inter-valley Hund's coupling is neglected, these two states are degenerate due to $SU(2)_+ \times SU(2)_-$ symmetry — the SDW may be obtained from the CDW by applying a spin-rotation in one valley only. Within Hartree-Fock, we find that the mean-field order parameter corresponds to one of these two phases (see Fig. 6 in the supplement).

Based on the symmetries of the original Hamiltonian, other kinds of IVC states are also possible: (i) An IVC which spontaneously breaks time-reversal symmetry in the orbital sector. Such a state has transverse currents on bonds, corresponding to a 3-sublattice orbital antiferromagnet (also called a current density wave). In absence of inter-valley Hund's coupling, this state is degenerate with a 3 sublattice spin-flux density wave, which may be obtained by a spin-rotation in one valley only. (ii) An IVC which also breaks C_3 symmetry, i.e., a nematic IVC state. We do not find evidence for either \mathcal{T} -breaking in the orbital sector, or C_3 breaking in our numerics, thus excluding the above phases.

The remaining possibility is for the IVC order parameter to have additional incommensurate momentum \mathbf{q} , i.e., within mean-field theory we have a non-zero expectation value for $n_{ss'}^{IV}(\mathbf{q}) = \langle c_{+,k,s}^\dagger c_{-,k+\mathbf{q},s'} \rangle \neq 0$ for $\mathbf{q} \neq 0$. Such states would correspond to incommensurate versions of the density-wave states previously discussed. However, the IVC susceptibility computed by You and Vishwanath in Phys. Rev. B 105, 134524 (2022) and reproduced below for convenience, indicates that it is peaked at $\mathbf{q} = 0$. Thus, we do not expect the IVC order parameter to condense at an incommensurate momentum \mathbf{q} , and restrict ourselves to the simplest ($\mathbf{q} = 0$) IVC state.

Can the authors elucidate the e-p couplings and explain why BCS superconductivity should be excluded?

To be clear, we are certainly not claiming that superconductivity due to electron-phonon interactions can be excluded. In fact, an acoustic phonon-mediated pairing scenario has been investigated in Chou *et al*, Phys. Rev. Lett. 127, 187001 (2021). In a recent preprint (arXiv:2206.01213), it is shown that the coupling between optical phonons and electrons near charge neutrality is very weak in RTG, due to the sublattice polarized nature of the wavefunctions. In addition, there are several experimental evidences that make the investigation of an alternative scenario worthwhile.

First, the pertinent experimental observation is that superconductivity in RTG always appears proximal to a symmetry-broken phase, indicating that order parameter fluctuations may have an important role to play in its origin. Second, phonons would act as an external bath for electrons and lead to a strong linear in T resistivity in the metallic state above the Bloch–Grüneisen temperature, which has not been observed in RTG. Finally, acoustic phonons do not choose between a singlet and a triplet superconductor, as the phonon-mediated interactions are fully $SU(2)_+ \times SU(2)_-$ symmetric. Suppose we could characterize the phase diagram by a single Hund's coupling J_H . Then, the presence of spin-polarized, valley-unpolarized phases in the phase diagram indicates that J_H is ferromagnetic. In such a scenario, a superconducting mechanism based solely on phonons would always predict a spin-triplet superconductor, in contradiction with experiments. Our proposed scenario can explain both the presence of spin-polarized phases and spin-singlet superconductivity within a single, consistent picture.

In summary, we thank the Referees for carefully reading our manuscript and their insightful comments. We believe we have addressed all issues raised with our additional computations and clarifications in the revised version, and thank the Referees for helping us to improve the manuscript.

List of changes made:

1. Added Hartree-Fock phase diagram (Fig. 2(b)) as a function of hole-doping and displacement field.
2. Added justification for using near-critical fluctuations of the IVC order parameter as pairing glue, by performing a detailed computation of the correlation length using Hartree-Fock (appendix F4).
3. Discussed the lack of time-reversal and C_3 breaking in Hartree-Fock to rule out orbital loop-current states and nematic IVC order.
4. Added rationale for projection of interaction onto valence band in Appendix F2.
5. Referred to previous theoretical work on superconductivity in R-graphite.
6. Edited minor typos pointed out by the referees.
7. Updated published references.

All changes in text are marked with red in the revised manuscript.

Reviewers' Comments:

Reviewer #2:

Remarks to the Author:

I have read through the response letter and the revised manuscript. In my opinion, the authors have thoroughly addressed all comments and improved the manuscript. Therefore, I think that the manuscript is suitable for publication. I have a minor comment.

The authors should check Eqs. (B1) and (F28), which look inconsistent to me.

Reviewer #3:

Remarks to the Author:

The authors have professionally answered my questions. I am happy to recommend its publication as is.

If possible, I suggest adding those valuable discussions in the response letter to the discussions sections of the paper. In other words, to highlight the pros and cons of the specific approach used by the authors could be equally valuable to readers who are interested in the studied topic in the long run.

Point-by-point response to referee 2:

I have read through the response letter and the revised manuscript. In my opinion, the authors have thoroughly addressed all comments and improved the manuscript. Therefore, I think that the manuscript is suitable for publication.

We thank the referee for their positive evaluation of the revised manuscript, and for recommending publication.

I have a minor comment. The authors should check Eqs. (B1) and (F28), which look inconsistent to me.

We thank the referee for pointing out the inconsistency between Eq. (B1) and Eq. (F28). In response to a previous referee comment, we changed the convention for $\chi_{\rho\rho}$ in Eq. (B1), but inadvertently forgot to change it in Eq. (F28). We have fixed Eq. (F28) and the subsequent text in the revised version.

Point-by-point response to referee 3:

The authors have professionally answered my questions. I am happy to recommend its publication as is.

We thank the referee for their appreciative comment about the work, and for recommending publication as is.

If possible, I suggest adding those valuable discussions in the response letter to the discussions sections of the paper. In other words, to highlight the pros and cons of the specific approach used by the authors could be equally valuable to readers who are interested in the studied topic in the long run.

We thank the referee for this valuable suggestion, and have highlighted the pros and cons of our approach by adding the following text to the discussion section.

However, acoustic phonons do not choose between a singlet and a triplet superconductor, as the phonon-mediated interactions are fully $SU(2)_+ \times SU(2)_-$ symmetric (optical phonons do not preserve this symmetry, but coupling of low-energy electrons to optical phonons is very weak in RTG under strong displacement fields). Suppose we could characterize the phase diagram by a single Hund's coupling J_H . Then, the presence of spin-polarized, valley-unpolarized phases in the phase diagram indicates that J_H is ferromagnetic. In such a scenario, a pairing mechanism based solely on acoustic phonons would always predict a spin-triplet superconductor, in contradiction with the experimental observation for SC1. Our proposed scenario can explain both the presence of spin-polarized phases and spin-singlet superconductivity within a single, consistent picture.

While our phenomenological treatment of coupling between electrons and soft-modes only allows us obtain an estimate of the superconducting critical temperature, our work motivates numerical explorations to determine T_c accurately as a function of carrier density and electric field in RTG.

In summary, we thank the Referees for their positive evaluation of our manuscript, and their critical comments. We believe we have addressed all the comments in the revised version, which should be fit for publication.

List of changes made:

1. Fixed a sign in Eq. (F28) and the text following immediately afterwards.
2. Added comments on the pros and cons of our approach in the discussion section, as recommended by referee.
3. Added a reference to justify the weak coupling of electrons to optical phonons.
4. Added brief title summarizing figure to figure legends, as per Nature Communications formatting guide.
5. Edited a typo in the supplementary information for typical speed of IVC soft modes.
6. Made minor changes to the acknowledgements section.
7. Placed acknowledgements section after references, and added statement of author contributions and competing interests, to fit formatting guide for Nature Communications.

All changes in the main text are marked with red in the revised manuscript.